# Increased Omega-3 Fatty Acid Intake Is Inversely Associated with Subclinical Inflammation in Healthy Elderly Men, Based on the 2015–2018 Korean National Health and Nutrition Examination Survey

**DOI:** 10.3390/nu13020338

**Published:** 2021-01-24

**Authors:** Woojung Yang, Jong Hun Lee, Jae-woo Lee, Yonghwan Kim, Ye-Seul Kim, Hyo-Sun You, Hee-Taik Kang

**Affiliations:** 1Department of Family Medicine, Chungbuk National University Hospital, Cheongju 28644, Korea; kineto@naver.com (W.Y.); shrimp0611@gmail.com (J.-w.L.); airsantajin@gmail.com (Y.K.); yesul86@naver.com (Y.-S.K.); hyo920@gmail.com (H.-S.Y.); 2Department of Food Science and Biotechnology, Gachon University, Seongnam 13120, Korea; foodguy@gachon.ac.kr; 3Department of Family Medicine, Chungbuk National University College of Medicine, Cheongju 28644, Korea

**Keywords:** omega-3 fatty acids, C-reactive protein, subclinical inflammation, cardiovascular disease, omega-3 fatty acid ratio

## Abstract

(1) Background: Subclinical inflammation as a risk factor of cardiovascular diseases was clinically measured using C-reactive protein (CRP) level. (2) Methods: This study was cross-sectionally designed based the 2015–2018 Korean National Health and Nutrition Examination Survey (KNHANES). The ratio of daily omega-3 fatty acids to energy intake (ω3FA ratio) was classified into four quartile groups (Q1, <0.3%; Q2, 0.3%–<0.6%; Q3, 0.6%–<1.0%; and Q4, ≥1.0% in both sexes). Logistic regression analysis was conducted to investigate the association between the ω3FA ratio and subclinical inflammation defined as CRP levels ≥3 mg/dL. (3) Results: The ω3FA ratio in subjects without and with subclinical inflammation was 0.8% and 0.7% in men (*p*-value = 0.001), and 0.8% and 0.8% in women (*p*-value = 0.491), respectively. The prevalence of subclinical inflammation in males decreased with increasing quartile of ω3FA ratio (12.9%, 9.6%, 7.4%, and 7.7%, *p*-value = 0.033), while female prevalence was not significant among quartile groups. Compared to Q1, odds ratios (95% confidence intervals) for subclinical inflammation of Q2, Q3, and Q4 were 0.740 (0.465–1.177), 0.564 (0.341–0.930), and 0.549 (0.317–0.953) in males, and 1.066 (0.653–1.741), 1.105 (0.600–1.718), and 0.934 (0.556–1.571) in females after full adjustment. (4) Conclusion: The ω3FA ratio is associated with subclinical inflammation in men.

## 1. Introduction

Because half of all myocardial infarctions occur in healthy people [1], prevention of cardiovascular diseases in healthy people is emerging as an important healthcare issue, especially in elderly people who have an inevitable cardiovascular disease risk factor—age. However, because most healthy people are likely to give little attention to their health regardless of age, the best way to prevent cardiovascular diseases in healthy elderly people is simple lifestyle modification.

Omega-3 fatty acids (ω3FAs) are macronutrients abundant in marine fishes. Several mechanisms by which ω3FAs can reduce cardiovascular disease risk are understood, including triglyceride reduction, vasodilatory nitric oxide (NO) increase [2], and thrombogenic thromboxane reduction [3]. Accordingly, ω3FA intake is recommended for prevention of cardiovascular disease in people with a history of cardiovascular diseases [4].

However, the protective effects of ω3FAs in healthy populations remain a topic of controversy. The American Heart Association announced that the recommendation of ω3FAs for healthy people is not backed by sufficient evidence [5], although general consumption of ω3FAs was once the formal recommendation of the organization. The 2016 European Society of Cardiology and European Atherosclerosis Society guidelines for prevention of cardiovascular disease has also stated that this issue is debatable. 

On the other hand, it was recently reported that high-sensitivity C-reactive protein (hsCRP) can predict the risk of cardiovascular events in any subject, including subjects from a healthy population, combined with levels of cholesterols, triglycerides, and glucose [1,6,7]. This is because chronic low-grade inflammation, also known as subclinical chronic inflammation, plays an important role in atherosclerosis, and hsCRP precisely and reliably reflects this condition [1]. 

The purpose of this study is to evaluate whether the ratio of daily ω3FA consumption to daily energy intake is associated with subclinical inflammation as a risk factor of cardiovascular diseases in healthy elderly people in Korea, using hsCRP level, based on the 2015–2018 Korean National Health and Nutrition Examination Survey (KNHANES).

## 2. Materials and Methods 

### 2.1. Study Population

Under the National Health Promotion Act, the KNHANES, a Korean nationwide representative cross-sectional survey, has been conducted since 1998 to evaluate and monitor the nutritional and health status of Koreans living in Korea by the Korea Disease Control and Prevention Agency (KDCA). The KNHANES collects information by staged, stratified, clustered, and systematic probability sampling based on sex, age, and geographic area using household registries to be representative of the whole Korean population residing in Korea. Up to the present, the KNHANES has been administered in seven phases: KNHANES phases I (1998), II (2001), III (2005), IV (2007–2009), V (2010–2012), VI (2013–2015), and VII (2016–2018). Among these, we first chose data from KNHANES 2018 (the most recent) to provide up-to-date health statistics. However, the range of data needed to date back to 2015 (the oldest survey that includes hsCRP values) to achieve statistical power, which is inevitably decreased by exclusion criteria for selection of healthy older adults.

The KNHANES consists of three components: a health interview, a health examination, and a nutrition survey. Trained staff including physicians conduct health interviews and health examinations at mobile examination centers, and nutrition surveys are followed up by dieticians’ visits to the homes of participants [8,9]. Through these processes, the KNHANES gathers information from respondents regarding health status, health behaviors, socioeconomic status, laboratory test results, and nutritional status. 

Using the dataset from 2015 to 2018, to investigate the association between ω3FA intake and subclinical inflammation in healthy older adults (removing the potential for increased C-reactive protein (CRP) level due to other causes), we excluded participants with hsCRP level over 10 mg/L, which is the value suggesting the presence of acute inflammation [1,10,11]. In addition, participants with histories of any malignancy or other chronic morbidities such as diabetes mellitus, coronary heart disease, renal failure, and severe rheumatoid arthritis were excluded. Lastly, we excluded participants younger than 60 years. A total of 4804 participants (2028 men and 2776 women) was included and analyzed in the final study (Figure 1). 

### 2.2. Definitions of High Cardiovascular Disease Risk Subclinical Inflammation and ω3FA Ratio

Subclinical inflammation is defined as an hsCRP level higher than 3 and equal to or less than 10 mg/L [10]. According to the American Heart Association and the U.S. Centers for Disease Control and Prevention, high cardiovascular disease risk is associated with subclinical inflammation in which CRP level in healthy adults is higher than 3 mg/dL [1,10,11].

The amount of daily ω3FA and calorie intake was estimated after analyzing the 24 h recall methods using Can-Pro nutrient intake assessment software that was developed by the Korean Nutrition Society. The ratio of daily ω3FA intake to energy intake (ω3FA ratio (unit, %)) was calculated by dividing ω3FA (unit, Kcal/day) by calorie intake (unit, Kcal/day), and multiplying them by 100. For analysis, the ratio of daily ω3FA intake to energy intake was categorized into four quartile groups in both sexes as follows: Q1, <0.3%; Q2, 0.3%–<0.6%; Q3, 0.6%–<1.0%; and Q4, ≥1.0%. Hereafter, we will call the ratio of daily ω3FA intake to energy intake the ω3FA ratio.

### 2.3. Definitions of Other Variables

Sufficient physical activity was defined as engaging in moderate physical activity for a total period of more than 150 min per week or engaging in vigorous physical activity for a total period of more than 75 min per week. Men who reported drinking more than seven alcoholic beverages and women who reported drinking more than five alcoholic beverages more than twice a week were designated as heavy alcohol drinkers [12]. Occupation was categorized into three groups of (1) manual workers (clerks; service and sales workers; skilled agricultural, forestry, and fishery workers; persons who operate or assemble crafts, equipment, or machines; and elementary workers), (2) office workers (general managers, government administrators, professionals, and simple office workers), and (3) other (unemployed persons, housekeepers, and students). Educational status was classified into four groups by total years of education: <6 years, 6–<9 years, 9–<12 years, and ≥12 years of education. Marital status was divided into two categories of (1) married and not separated (referring to people who were married and living together without separation) and (2) single people who were unmarried, separated, divorced, or widowed. Blood pressure (BP) was measured three times in subjects using a standard mercury sphygmomanometer (Baumanometer; Baum Co., Inc., Copiague, NY, USA), and a mean value of the last two measurements was defined as the BP of subjects. Aspartate transaminase (AST), total cholesterol, and glucose were measured by the International Federation of Clinical Chemistry (IFCC) techniques without pyridoxal-5-phosphate (P5P), an enzymatic method, and hexokinase ultraviolet, respectively (Hitachi 7600 Automatic Analyzer; Hitachi, Ltd., Tokyo, Japan). High-sensitivity C-reactive protein (hsCRP) values were measured by immunoturbidimetry (Cobas analyzer; F. Hoffmann-La Roche Ltd., Basel, Switzerland). When the hsCRP value from 2016 to 2018 was greater or less than the detection range (0.00–20.00 mg/L in 2016, 0.15–20.00 mg/L in 2017 and 2018), we estimated the value at 20.01 or 0.14 mg/L based on guidelines adopted in the 2015 KNHANES.

### 2.4. Assessment of Daily Nutrient Intake

Daily nutrient intake was assessed using the 24 h recall method, which was conducted by trained dietitians. After all participants were instructed to maintain their usual dietary habits before the nutrition survey, they were asked to recall the foods and amounts that they had consumed in the last 24 h. Final daily intake of nutrient was calculated from the information recorded on the Food Intake Questionnaire by eye measurement, using the CAN-Pro nutrient intake assessment software program developed by the Korean Nutrition Society.

### 2.5. Statistical Analysis

All data on continuous variables are presented as means ± standard errors (SEs). Data on categorical variables are presented as percentages ± SEs. All sampling and weight variables were stratified by sex. Statistical software SAS version 9.4 (SAS Institute Inc., Cary, NC, USA) was utilized for statistical analysis to process the complicated sampling design and to provide nationally representative estimates. Survey regressions (an SAS version 9.4 syntax) and chi-squared (χ^2^) tests were used to compare sexes, quartiles of ω3FA ratio, and subjects with or without subclinical inflammation. We also estimated adjusted odds ratios (ORs) and 95% confidence intervals (CIs) by multivariate logistic regression models to investigate factors associated with subclinical inflammation according to ω3FA ratio. p-values were calculated by multiple logistic regression analyses with weighting of the survey design (adjusted with age, body mass index, total cholesterol, systolic BP, fasting plasma glucose, AST, smoking status, alcohol intake, economic status, marital status, education duration, occupation, sufficient physical activity, and protein intake). All statistical tests were two-tailed, and statistical significance was considered at *p*-values < 0.05.

## 3. Results

Table 1 shows participant baseline characteristics according to sex. The average ages of males and females were 69.3 and 69.1 years, respectively (*p*-value = 0.549). Male and female daily energy intakes were 2041.7 and 1556.0 kilocalories (kcals), respectively (*p*-value < 0.001). Males consumed more macronutrients (carbohydrates, proteins, and fats) than females (all *p*-values < 0.001). However, in terms of the ω3FA ratio, both sexes consumed the same total percentage (0.8%) (*p*-value = 0.661).

Table 2 demonstrates population characteristics according to the quartile of ω3FA ratio. In males, average age was significantly different among the four quartile groups (Q1: 70.4, Q2: 68.8, Q3: 69.2, and Q4: 68.7, *p*-value = 0.001). Protein intake increased with ω3FA ratio quartile, and daily energy intake tended to increase with ω3FA ratio quartile (all *p*-values < 0.001). On the other hand, the number of current smokers decreased with higher ω3FA ratio quartile (*p*-value = 0.011) in males. In females, higher ω3FA ratio quartile groups were younger than lower ω3FA ratio quartile groups (*p*-value < 0.001). Daily energy intake and protein intake were positively associated with ω3FA ratio quartile (all *p*-values < 0.001). Significant differences among the four groups were not observed for current smokers (*p*-value = 0.298), while higher ω3FA ratio quartile groups were more likely in females with higher economic status, who were married and not separated, and who had a higher education status (all *p*-values < 0.001).

Table 3 presents fat intake according to the presence of subclinical inflammation. In males, the ω3FA ratio in subjects without subclinical inflammation was significantly higher than that in subjects with subclinical inflammation (0.8% in subjects with no subclinical inflammation vs. 0.7% in subjects with subclinical inflammation, *p*-value = 0.001). However, ω3FA ratio in females was not significantly different between the two groups with and without subclinical inflammation (*p*-value = 0.491). Omega-6/omega-3 ratios were not significantly different in both sexes (all *p*-values > 0.05).

Figure 2 demonstrates the prevalence of subclinical inflammation according to quartile of ω3FA ratio. The prevalence of subclinical inflammation in males decreased with increasing quartile of ω3FA ratio (12.9%, 9.6%, 7.4%, and 7.7%, respectively, *p*-value = 0.033). In contrast, there was no significant difference in the prevalence of subclinical inflammation among female quartile groups (*p*-value = 0.958). 

Table 4 presents logistic regression models according to ω3FA ratio quartile. Compared to Q1, ORs (95% CIs) for subclinical inflammation of Q2, Q3, and Q4 of ω3FA ratio were 0.754 (0.489–1.162), 0.561 (0.356–0.883), and 0.595 (0.368–0.962), respectively, in males, and 1.030 (0.637–1.666), 1.024 (0.620–1.694), and 0.972 (0.600–1.575), respectively, in females after adjustment for age. After full adjustment, including for omega-6/omega-3 ratio, ORs (95% CIs) of Q2, Q3, and Q4 of ω3FA ratio were 0.740 (0.465–1.177), 0.564 (0.341–0.930), and 0.549 (0.317–0.953), respectively, in males, and 1.066 (0.653–1.741), 1.105 (0.600–1.718), and 0.934 (0.556–1.571), respectively, in females.

## 4. Discussion

This study found that the ω3FA ratio was inversely associated with the prevalence of subclinical inflammation in apparently healthy elderly males (but not in females), even after adjusting for confounding variables.

Cardiovascular diseases are the second most common cause of death in Korea [13]. They cause fatal complications to the heart itself and other organs even if patients survive. Cardiovascular diseases also result in an economic burden (such as costs for treatment and maintenance of patients’ daily lives) to patients, their caregivers, and the wider society [14]. Therefore, prevention is the best way to decrease losses from cardiovascular diseases and their comorbidities.

In atherosclerosis, a pathogenesis of cardiovascular diseases, inflammation plays a greater role in initiation and progression to vulnerable atherosclerotic plaque than do lipids in the blood [15,16,17,18,19]. Oxidized low-density lipoproteins (LDLs), which are increased in dyslipidemia; free radicals formed by smoking; and the shearing stress of blood increased in hypertension damage endothelial cells. In this way, a chronic low-grade inflammatory process, also known as subclinical chronic inflammation, is initiated on vessel walls [16,17,18]. As a result of inflammation, endothelial cells attract macrophages and enhance their activities, and atherosclerotic plaque is formed by interactions among these cells within vessels [15]. Furthermore, as inflammatory processes continue, inflammatory cells such as macrophages keep infiltrating on plaque and secrete matrix metalloproteinases [15]. Ultimately, atherosclerotic plaques become vulnerable and rupture, producing cardiovascular events [15]. This pathogenesis of cardiovascular disease demonstrates how CRP, a biomarker reflecting the extent of inflammation [1], may predict the risk of cardiovascular events in all subjects, including apparently healthy adults [6,20,21]. Furthermore, these patterns explain how CRP can predict the risk of cardiovascular diseases even in individuals with normal lipid levels [1,7]. 

After recognizing the value of CRP in predicting cardiovascular diseases, its proper function began to draw attention from many researchers. CRP possesses proatherogenic properties similar to those of the inflammation process and, thereby, can influence the pathogenesis of cardiovascular diseases. CRP attracts macrophages and helps them adhere to endothelial injury sites by stimulating endothelial cells to secrete monocyte chemoattractant protein 1 (MCP-1), intercellular adhesion molecule 1 (ICAM-1), and vascular cell adhesion molecule 1 (VCAM-1) [22,23]. In addition, CRP activates macrophages in atherosclerotic plaques to express extracellular matrix metalloproteinase inducer and matrix metalloproteinase-9 [24].

On the other hand, ω3FAs are known to have anti-inflammatory properties that suppress inflammation throughout human bodies, including the vascular environment [25,26]. ω3FAs lower the expression of adhesion molecules on endothelial cells [15]. For instance, expression of ICAM-1 and VCAM-1 on endothelial cells is reduced by ω3FAs [27,28]. Furthermore, ω3FAs decrease the expression of chemoattractants of macrophages such as MCP-1 on endothelial cells. Through these effects, ω3FAs also decrease infiltration of inflammatory cells such as macrophages, which secrete matrix metalloproteinases into atherosclerotic plaques, stabilize atherosclerotic plaques, and prevent cardiovascular events [15,29]. Therefore, ω3FAs may reduce the risk of cardiovascular diseases and the CRP level through their anti-inflammatory effects even in apparently healthy people.

ω3FAs are also found to directly reduce the CRP level by suppressing the expression of toll-like receptor 4 (TLR4), a key modulator of pro-inflammatory cytokines including interleukin-6 (IL-6) [30]. Although the precise pathway from TLR4 to IL-6 remains unknown [30], previous studies have shown that ω3FAs decrease the level of IL-6, which increases the rate of transcription of the *CRP* gene and the translation of CRP mRNA, even in endothelial cells [15,26,31,32]. 

However, in terms of cardiovascular disease mortality, the evidence for ω3FA intake reducing cardiovascular disease mortality is known to be insufficient even in patients with high cardiovascular disease risk, at this point in time [33]. Therefore, caution is needed when expanding the implications of our findings to other related health outcomes including cardiovascular disease mortality.

Many previous studies have shown that ω3FAs are associated with CRP level. However, most previous studies utilize CRP level as a sensitive marker reflecting the degree of systemic inflammation to demonstrate the anti-inflammatory effects of ω3FAs. Through this approach, previous research presents the beneficial effects of ω3FAs on inflammation only at the laboratory level without relating to health outcomes. The current study expands these effects to the clinical level in terms of the risk of cardiovascular diseases in apparently healthy older adults using the category of high cardiovascular disease risk: subclinical inflammation.

A noteworthy point out of our findings is that there is a significant association between ω3FA ratio and the prevalence of subclinical inflammation in older men but not in women. In our study, men had significantly higher CRP levels. Furthermore, in connection with the ω3FA ratio, men’s CRP levels were so high that the level of men’s Q4 was equal to that of women’s Q1. The cause of this difference seems to come from higher risk factors for high CRP that men had. In this study, men’s percentages of current smoking and heavy alcohol drinking were significantly higher than those of women. With this difference in CRP level between sexes, ω3FA intake seems to have different effects on subclinical inflammation in this study, depending on sex. In other words, because men had a significantly higher CRP level than women in this study, we assumed that men had greater potential for ω3FA intake to decrease CRP level. On the other hand, previous studies have shown inconsistent results when it comes to the association or causality between ω3FA consumption and CRP level in women, depending on the study design, dosage, or duration of ω3FA intake [26].

Several limitations should be considered in interpreting the findings of this study. First, because hsCRP values out of the detection range were approximated to the closest values, the CRP values used in our statistical analysis may be slightly different from actual levels. Second, we cannot conclude causality from this study because it was cross-sectional in design. Third, because we gathered data about ω3FA intake by a 24 h recall method instead of with actual measurements of ω3FAs in the blood or red blood cell membranes of subjects, we cannot exclude the potential for recall or reporting bias. However, if taking measures to be representative of participants’ usual dietary habits excluding day-to-day variation and to reduce measurement variability depending on interviewers, it can be used as one of the reliable ways to analyze nutrients intake. Actually, some studies have shown that the 24 h recall method could be a sufficient method for assessing dietary intake after taking measures including the aforementioned [34]. With this background, many nation-wide nutrition surveys including the National Health and Nutrition Examination Survey (NHANES) adopted the 24 h recall method as a reliable way to assess nutrient intake. Fourth, the information on ω3FA intake through dietary supplements was not included in the analyses because the KNHANES had not gathered the information on ω3FA supplements. Therefore, we were not able to completely exclude the confounding effects of ω3FA supplements in this study.

Despite these limitations, this study has several strengths that distinguish it from existing research. First, this study utilizes a population-based sample selected by an established survey design to produce estimates that accurately represent the Korean population. Second, to the best of our knowledge, this is the first Korean study to evaluate the association of ω3FA intake and ω3FA ratio with the prevalence of high cardiovascular disease risk subclinical inflammation in healthy older adults using nationally representative data. Third, to minimize the confounding effects of several factors, especially protein levels that increase simultaneously with CRP level when marine fishes are consumed, we analyzed the data not only using ω3FA ratio instead of amounts of ω3FAs but also by adjusting for daily protein intake.

## 5. Conclusions

This study shows that increased ω3FA-ratio-based intake is associated with decreased subclinical inflammation in apparently healthy elderly males. However, longitudinally designed cohort studies or interventional studies are further needed to confirm whether the higher ratio of ω3FA intake to daily calorie consumption decreases subclinical inflammation in healthy older adults to result in the prevention of cardiovascular events.

## Figures and Tables

**Figure 1 nutrients-13-00338-f001:**
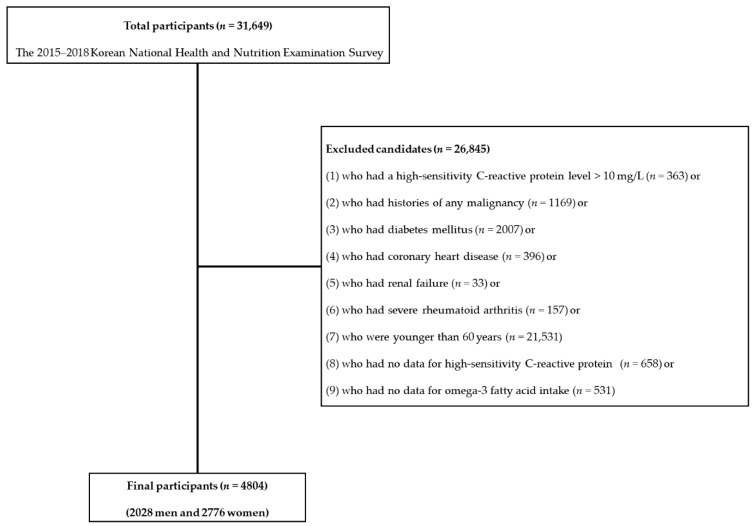
Flowchart of exclusion criteria.

**Figure 2 nutrients-13-00338-f002:**
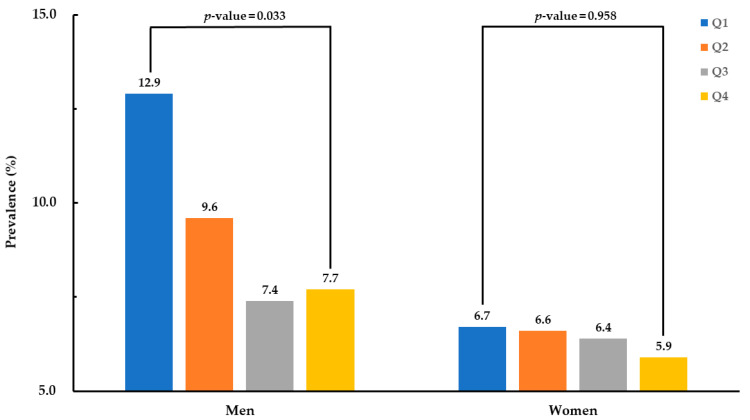
Prevalence of subclinical inflammation according to quartile group of omega-3 fatty acid ratio.

**Table 1 nutrients-13-00338-t001:** Population characteristics according to sex.

Characteristic	Males	Females	*p*-Value
Number	2028	2776	
Age, years	69.3 ± 0.2	69.1 ± 0.2	0.549
BMI, kg/m^2^	23.9 ± 0.1	24.3 ± 0.1	<0.001 *
Energy intake, Kcal/day	2041.7 ± 21.5	1556.0 ± 15.2	<0.001 *
Carbohydrate intake, g/day	333.3 ± 3.6	275.2 ± 2.8	<0.001 *
Protein intake, g/day	69.3 ± 0.9	51.3 ± 0.6	<0.001 *
Fat intake, g/day	34.5 ± 0.7	26.1 ± 0.5	<0.001 *
ω3FA ratio, %	0.8 ± 0.0	0.8 ± 0.0	0.661
Omega-6/omega-3 ratio	6.3 ± 0.2	6.4 ± 0.1	0.807
hsCRP, mg/L	1.2 ± 0.0	1.0 ± 0.0	<0.001 *
Total cholesterol, mg/dL	187.4 ± 1.0	199.6 ± 0.9	<0.001 *
Systolic blood pressure, mmHg	126.9 ± 0.4	127.7 ± 0.4	0.157
Fasting plasma glucose, mg/dL	102.9 ± 0.5	99.8 ± 0.3	<0.001 *
AST, IU/L	24.8 ± 0.3	23.6 ± 0.2	0.002 *
Current smoking, %	22.3 ± 1.1	2.7 ± 0.4	<0.001 *
Heavy alcohol intake, %	12.7 ± 0.9	1.2 ± 0.2	<0.001 *
Economic status, %			<0.001 *
Low	30.4 ± 1.3	38.9 ± 1.2	
Middle–low	28.3 ± 1.1	27.2 ± 1.0	
Middle–high	22.3 ± 1.1	19.3 ± 1.0	
High	19.1 ± 1.1	14.6 ± 1.0	
Marital status, %			<0.001 *
Married and not separated	89.7 ± 0.8	58.7 ± 1.2	
Single	10.3 ± 0.8	41.3 ± 1.2	
Education duration, %			<0.001 *
<6 years	34.7 ± 1.4	58.6 ± 1.3	
6–<9 years	18.2 ± 1.2	16.5 ± 0.8	
9–<12 years	26.2 ± 1.2	16.6 ± 0.9	
≥12 years	20.9 ± 1.2	8.2 ± 0.7	
Occupation, %			<0.001 *
Office workers	9.7 ± 0.8	2.4 ± 0.4	
Manual workers	42.3 ± 1.5	32.5 ± 1.1	
Other	48.1 ± 1.4	65.1 ± 1.1	
Sufficient physical activity, %	41.9 ± 1.4	34.8 ± 1.2	<0.001 *

All data are presented as mean ± standard errors (SEs) or percentage ± SEs. Abbreviations: BMI, body mass index; ω3FA, omega-3 fatty acid; ω3FA ratio, the ratio of daily ω3 fatty acid intake to energy intake; hsCRP, high-sensitivity C-reactive protein; AST, aspartate transaminase; IU/L, international units per liter. * *p*-value < 0.05. *p*-values were calculated from survey regressions.

**Table 2 nutrients-13-00338-t002:** Population characteristics according to quartile of ω3FA ratio.

Males	Q1	Q2	Q3	Q4	*p*-Value
	(<0.3%)	(0.3%–<0.6%)	(0.6%–<1.0%)	(≥1.0%)	
Number	538	498	490	502	
Age, years	70.4 ± 0.4	68.8 ± 0.3	69.2 ± 0.4	68.7 ± 0.3	0.001 *
BMI, kg/m^2^	23.8 ± 0.2	23.7 ± 0.1	23.9 ± 0.1	24.0 ± 0.1	0.315
Energy intake, Kcal/day	1873.6 ± 43.1	2093.2 ± 36.6	2090.1 ± 41.9	2109.8 ± 38.6	<0.001 *
Carbohydrate intake, g/day	318.0 ± 6.9	352.6 ± 6.3	335.5 ± 7.0	326.9 ± 5.7	0.001 *
Protein intake, g/day	54.5 ± 1.5	69.2 ± 1.5	74.1 ± 1.6	79.3 ± 1.8	<0.001 *
Fat intake, g/day	24.2 ± 1.1	33.1 ± 1.2	36.1 ± 1.2	44.4 ± 1.5	<0.001 *
ω3FA ratio, %	0.2 ± 0.0	0.5 ± 0.0	0.8 ± 0.0	1.9 ± 0.1	<0.001 *
Omega-6/omega-3 ratio	10.7 ± 0.6	6.7 ± 0.2	4.9 ± 0.1	3.0 ± 0.1	<0.001 *
hsCRP, mg/L	1.4 ± 0.1	1.2 ± 0.1	1.2 ± 0.1	1.1 ± 0.1	0.024 *
Total cholesterol, mg/dL	190.5 ± 1.9	186.1 ± 1.7	187.1 ± 1.8	186.1 ± 1.9	0.301
Systolic blood pressure, mmHg	128.7 ± 0.8	128.3 ± 0.8	125.9 ± 0.9	124.5 ± 0.8	0.001 *
Fasting plasma glucose, mg/dL	103.8 ± 1.0	103.7 ± 1.1	102.3 ± 1.1	101.8 ± 0.8	0.273
AST, IU/L	25.9 ± 0.6	24.1 ± 0.4	24.2 ± 0.4	25.0 ± 0.9	0.045 *
Current smoking, %	27.1 ± 2.2	24.0 ± 2.3	20.4 ± 2.2	17.6 ± 1.8	0.011 *
Heavy alcohol intake, %	16.0 ± 1.7	11.5 ± 1.7	13.0 ± 1.8	10.4 ± 1.5	0.077
Economic status, %					<0.001 *
Low	42.5 ± 2.4	29.7 ± 2.5	26.8 ± 2.3	22.5 ± 2.1	
Middle–low	27.1 ± 2.2	27.7 ± 2.2	26.4 ± 2.1	31.9 ± 2.4	
Middle–high	17.0 ± 1.9	21.8 ± 2.1	24.6 ± 2.1	25.6 ± 2.3	
High	13.4 ± 1.8	20.8 ± 1.9	22.2 ± 2.3	20.0 ± 2.0	
Marital status, %					0.001 *
Married and not separated	85.0 ± 1.8	88.6 ± 1.7	91.8 ± 1.5	93.2 ± 1.3	
Single	15.0 ± 1.8	11.4 ± 1.7	8.2 ± 1.5	6.8 ± 1.3	
Education duration, %					<0.001 *
<6 years	48.2 ± 2.7	36.7 ± 2.6	29.4 ± 2.2	24.9 ± 2.5	
6–<9 years	20.5 ± 2.2	17.7 ± 1.9	16.8 ± 1.9	17.7 ± 2.3	
9–<12 years	18.7 ± 2.0	25.0 ± 2.2	28.6 ± 2.4	32.3 ± 2.6	
≥12 years	12.7 ± 1.8	20.6 ± 2.1	25.1 ± 2.4	25.1 ± 2.2	
Occupation, %					0.471
Office workers	6.0 ± 1.1	10.7 ± 1.5	10.9 ± 1.7	10.9 ± 1.5	
Manual workers	46.1 ± 2.7	44.3 ± 2.6	39.6 ± 2.8	39.3 ± 2.5	
Other	47.9 ± 2.6	45.1 ± 2.7	49.5 ± 2.9	49.7 ± 2.6	
Sufficient physical activity, %	39.5 ± 2.7	44.1 ± 2.5	41.6 ± 2.8	42.4 ± 2.7	0.675
**Females**	**Q1**	**Q2**	**Q3**	**Q4**	***p*-Value**
	(<0.3%)	(0.3%–<0.6%)	(0.6%–<1.0%)	(≥1.0%)	
Number	710	695	682	689	
Age, years	70.7 ± 0.3	69.4 ± 0.3	68.7 ± 0.3	67.7 ± 0.3	<0.001 *
BMI, kg/m^2^	24.4 ± 0.1	24.3 ± 0.1	24.1 ± 0.1	24.1 ± 0.1	0.302
Energy intake, Kcal/day	1438.4 ± 30.0	1546.2 ± 28.6	1572.1 ± 28.0	1667.2 ± 28.9	<0.001 *
Carbohydrate intake, g/day	277.8 ± 5.9	281.0 ± 5.3	271.9 ± 5.0	270.0 ± 4.8	0.388
Protein intake, g/day	40.0 ± 1.1	49.3 ± 1.0	55.1 ± 1.1	60.8 ± 1.3	<0.001 *
Fat intake, g/day	16.1 ± 0.7	22.7 ± 0.7	28.2 ± 0.8	37.4 ± 1.1	<0.001 *
ω3FA ratio, %	0.2 ± 0.0	0.4 ± 0.0	0.8 ± 0.0	1.9 ± 0.0	<0.001 *
Omega-6/omega-3 ratio	10.4 ± 0.4	6.6 ± 0.1	5.3 ± 0.1	3.2 ± 0.1	<0.001 *
hsCRP, mg/L	1.1 ± 0.1	1.1 ± 0.1	1.0 ± 0.1	1.0 ± 0.1	0.251
Total cholesterol, mg/dL	198.7 ± 1.7	199.4 ± 1.7	198.4 ± 1.7	201.9 ± 1.8	0.507
Systolic blood pressure, mmHg	129.2 ± 0.9	128.1 ± 0.8	127.2 ± 0.8	126.2 ± 0.8	0.074
Fasting plasma glucose, mg/dL	101.6 ± 0.8	98.6 ± 0.6	99.9 ± 0.7	99.0 ± 0.6	0.012 *
AST, IU/L	23.6 ± 0.4	23.7 ± 0.4	23.8 ± 0.6	23.3 ± 0.3	0.851
Current smoking, %	2.3 ± 0.6	3.8 ± 1.0	1.8 ± 0.5	3.0 ± 0.9	0.298
Heavy alcohol intake, %	0.8 ± 0.3	1.6 ± 0.6	1.1 ± 0.4	1.2 ± 0.4	0.737
Economic status, %					<0.001 *
Low	49.1 ± 2.3	40.0 ± 2.2	34.3 ± 2.3	32.0 ± 2.1	
Middle–low	27.9 ± 2.1	26.1 ± 1.9	25.1 ± 2.0	29.5 ± 2.0	
Middle–high	15.7 ± 1.9	19.7 ± 1.8	20.9 ± 2.0	21.0 ± 1.8	
High	7.3 ± 1.1	14.2 ± 1.7	19.6 ± 2.0	17.5 ± 1.8	
Marital status, %					<0.001 *
Married and not separated	50.7 ± 2.3	57.5 ± 2.1	60.3 ± 2.3	66.3 ± 2.2	
Single	49.3 ± 2.3	42.5 ± 2.1	40.0 ± 2.3	33.7 ± 2.2	
Education duration, %					<0.001 *
<6 years	72.3 ± 2.2	61.6 ± 2.2	51.8 ± 2.3	49.2 ± 2.3	
6–<9 years	13.5 ± 1.6	15.8 ± 1.6	18.0 ± 1.6	18.7 ± 1.8	
9–<12 years	10.2 ± 1.5	15.6 ± 1.7	20.7 ± 2.0	19.7 ± 1.9	
≥12 years	4.0 ± 1.0	7.0 ± 1.1	9.5 ± 1.4	12.3 ± 1.6	
Occupation, %					0.996
Office workers	0.9 ± 0.4	1.5 ± 0.7	2.9 ± 0.7	4.2 ± 0.9	
Manual workers	35.2 ± 2.3	34.8 ± 2.2	31.2 ± 2.0	29.1 ± 1.9	
Other	63.9 ± 2.3	63.7 ± 2.3	65.9 ± 2.1	66.7 ± 2.0	
Sufficient physical activity, %	30.3 ± 2.2	36.5 ± 2.2	35.5 ± 2.2	36.8 ± 2.3	0.122

All data are presented as mean ± SEs or percentage ± SEs. Abbreviations: ω3FA, omega-3 fatty acid; ω3FA ratio, ratio of daily omega-3 fatty acid intake to energy intake; BMI, body mass index; hsCRP, high-sensitivity C-reactive protein; AST, aspartate transaminase; IU/L, international units per liter. * *p*-value < 0.05. *p*-values were calculated from survey regressions.

**Table 3 nutrients-13-00338-t003:** Ratio of daily total fat and fatty acids intake to energy intake according to the presence of subclinical inflammation.

Ratio of Daily Total Fat and Fatty Acids Intake to Energy Intake (%)	Subclinical Inflammation	No Subclinical Inflammation	*p*-Value
Males, Number	197	1831	
Total fat	13.7 ± 0.6	14.6 ± 0.2	0.129
SFA	4.3 ± 0.2	4.4 ± 0.1	0.639
MUFA	4.1 ± 0.2	4.4 ± 0.1	0.282
PUFA	3.6 ± 0.2	4.2 ± 0.1	0.001 *
Omega-3 FA	0.7 ± 0.0	0.8 ± 0.0	0.001 *
Omega-6 FA	3.0 ± 0.1	3.4 ± 0.1	0.004 *
Omega-6/omega-3 ratio ^†^	6.8 ± 0.4	6.3 ± 0.2	0.229
Females, Number	184	2592	
Total fat	13.6 ± 0.7	14.5 ± 0.2	0.241
SFA	4.3 ± 0.2	4.4 ± 0.1	0.675
MUFA	4.1 ± 0.3	4.3 ± 0.1	0.430
PUFA	3.7 ± 0.2	4.3 ± 0.1	0.011 *
Omega-3 FA	0.8 ± 0.1	0.8 ± 0.0	0.491
Omega-6 FA	3.0 ± 0.1	3.4 ± 0.1	0.005 *
Omega-6/omega-3 ratio ^†^	6.2 ± 0.4	6.4 ± 0.1	0.752

All data are presented as mean (± standard errors). Abbreviations: SFA, saturated fatty acids; MUFA, monounsaturated fatty acids; PUFA, polyunsaturated fatty acids; ω3FA, ω3 fatty acids; ω6FA, ω6 fatty acids. * *p*-value < 0.05. *p*-values were calculated from survey regressions. ^†^ This variable does not represent the ratio of daily total fat and fatty acids intake to energy intake (%).

**Table 4 nutrients-13-00338-t004:** Adjusted odds ratios (ORs) and 95% confidence intervals (CIs) of subclinical inflammation by quartiles of ω3FA ratio.

Males	Q1	Q2	Q3	Q4
	(<0.3%)	(0.3%–<0.6%)	(0.6%–<1.0%)	(≥1.0%)
Number	538	498	490	502
Model 1	1 (ref)	0.754 (0.489–1.162)	0.561 (0.356–0.883)	0.595 (0.368–0.962)
Model 2	1 (ref)	0.752 (0.487–1.161)	0.555 (0.353–0.873)	0.590 (0.363–0.958)
Model 3	1 (ref)	0.740 (0.465–1.177)	0.564 (0.341–0.930)	0.549 (0.317–0.953)
**Females**	**Q1**	**Q2**	**Q3**	**Q4**
	(<0.3%)	(0.3%–<0.6%)	(0.6%–<1.0%)	(≥1.0%)
Number	710	695	682	689
Model 1	1 (ref)	1.030 (0.637–1.666)	1.024 (0.620–1.694)	0.972 (0.600–1.575)
Model 2	1 (ref)	1.096 (0.668–1.797)	1.094 (0.655–1.827)	1.051 (0.642–1.719)
Model 3	1 (ref)	1.066 (0.653–1.741)	1.105 (0.600–1.718)	0.934 (0.556–1.571)

Odds ratios (ORs) and 95% confidence intervals (CIs) were calculated using weighted multivariate logistic regression analyses. Model 1: Adjusted for age. Model 2: Adjusted for age, BMI, total cholesterol, systolic blood pressure, fasting plasma glucose, and AST. Model 3: Adjusted for current smoking status, heavy alcohol intake, economic status, marital status, education duration, occupation, sufficient physical activity, protein intake, and omega-6/omega-3 ratio in addition to variables of Model 2. Abbreviation: ω3FA, ω3 fatty acid; ω3FA ratio, ratio of daily ω3FA intake to energy intake; BMI, body mass index; AST, aspartate transaminase.

## Data Availability

For purposes of academic research, data from the Korean National Health and Nutrition Examination Survey (KNHANES) are available at no cost on the KNHANES website (http://knhanes.cdc.go.kr).

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
