# Peer review of "Increased Omega-3 Fatty Acid Intake Is Inversely Associated with Subclinical Inflammation in Healthy Elderly Men, Based on the 2015–2018 Korean National Health and Nutrition Examination Survey"

_nutrients, 2021, doi:10.3390/nu13020338_

Round 1

Reviewer 1 Report

Nutrients 1042718

Title: Increased omega-3 fatty acid intake is inversely associated with cardiovascular disease risk in healthy elderly men, based on the 2015-1018 Korean National Health and Nutrition Examination survey

This is a cross-sectional study using a population-based sample (n=4804) of healthy, elderly (>=60 years) Korean adults. Data were gathered between 2015-2018, as part of a national health and nutrition survey. The aim was to evaluate the association of the ratio of omega-3 fatty acids to daily energy intake with subclinical inflammation. Subclinical inflammation was defined via hsCRP circulating levels (3mg/L<hsCRP<=10mg/L). ω3FA ratio was calculated via a nutrition survey by a 24h recall method. Results showed that ω3FA ratio was associated with subclinical inflammation in men only.

The study is strong due to the large population-based sample. However, there are some major comments which need to be addressed.

Major comments:

Title:

Based on the primary aim of the study, the title is misleading. CVD risk was not measured as a variable. Instead, as stated in the primary aim of the study, subclinical inflammation was measured by the levels of hsCRP. Therefore the term “cardiovascular disease risk” should be retracted or changed to “subclinical inflammation”.

Materials and Methods:

Did participants sign informed consent?

Definition of healthy adults is missing.

Also, a statement on medication is missing. Participants on medication (i.e. for hypertention, for dyslipidemia, antiflammatories, other) were they considered as healthy or were they excluded? Inclusion/exclusion criteria are vague on this topic.

The method for the collection and calculation of ω3FAs is missing from the methods section. There is only a brief mention at the limitations section.

The fact that ω3FAs were measured solely by a 24h recall poses a serious limitation of the study. It would be useful if the authors provided the questionnaire pertaining to ω3FAs, perhaps as a supplementary material, or briefly discussed in the methods section.

Statistical analysis-Results:

Although multivariate models are necessary to identify independent predictors, it is also important to demonstrate the univariate associations for the variables included in the multivariate models.

Discussion:

The discussion on CVD and CRP is lengthy and introductory, lacking further critique and insight on the results. For example why was there an association for men but not women? Could it be because men had higher CRP to begin with? Or that they had much greater percentage of smoking and heavy drinking than women, implying thus greater subclinical inflammation

How do the authors interpret the OR results presented in Table 4 for both men and women.

Line 237-238: “ω3FAs can reduce the risk of cardiovascular diseases and CRP level through their anti-inflammatory effects” In general there is no solid evidence that this is true for healthy adults. However, ω3FA supplementation to prevent secondary CVD risk has been established. A broader discussion of the literature to present all aspects would be valuable.

A large part of the references is outdated (over 20years). Please include more up-dated references to convey current knowledge on ω3FA and hsCRP, as newer studies differentiate from older ones. For example older studies on ω3FAs showed a positive effect on CVD death risk reduction, but more recent studies don’t.

Lines 218-221: “CRP, a biomarker reflecting the extent of inflammation [1], can predict the risk of cardiovascular events in all subjects, including apparently healthy adults [6, 18, 19]”: This statement is too strong, because hsCRP is generally considered to be a weak index of subclinical inflammation as it does not necessarily reflect tissue inflammation. Please rephrase.

Minor

Line 117 “techniques without P5P” is this correct or should it rather be “with”?

Figure 1: provide the name of the y axis on the graph

Reviewer 2 Report

The article entitled “Increased Omega-3 Fatty Acid Intake is Inversely Associated with Cardiovascular Disease Risk in  Healthy Elderly Men, Based on the 2015-2018 Korean National Health and Nutrition Examination Survey” analyses the relation between daily omega-3 fatty acids (ω3FAs) consumption and subclinical inflammation in healthy elderly people using C-reactive protein levels.

This issue is interesting, however data about omega-6-omega-3 ratio is lacking.

Since the Importance of the Omega-6/Omega-3 Fatty Acid Ratio in Cardiovascular Disease (https://doi.org/10.3181/0711-MR-311), the authors should  also show the relation between subclinical inflammation and Omega-6/Omega-3 Fatty Acid Ratio both in men and women.

Minor comments:

Please explain the meaning of (ref) in Table 4

Tables 1-3:  for an easier reading, please indicate with asterisk the values statistically significant.

Reviewer 3 Report

The submitted article presents a topic with some relevance, however the way it was work does not semes to be the best. It has some lack of information, mainly in the results (eg description of the tables) and in the summary. The discussion also needs improvement. The points that need to be improved are described below:

Summary (line 20) - The methods need to be briefly described. When reading the summary, it is not possible to understand the type of study or the sample size. The results can be summarized so that the methods section can be properly described. Something that was not done.

Lines 96-98 - The autors should explain in more detail how was obtained the information on the dietary intake of participants and how the ratio was calculated, indicating units of omega 3 fatty acids intake.

Lines 181 to 185 - Table 3 has lack of information. What is the (n) of men and women with and without subclinical inflammation? How does the difference in the average values of the omega-3 ratios (0.1) have a greater statistical significance (p = 0.001) than the difference in the average values of the omega-6 ratios (with 0.4 difference and p = 0.004). The statistical test used to evaluate the differences between the two groups is not indicated in the text or at the table bottom.

Lines 189 to 197 - There is no information about the sample size in men and women used to apply the models. This information (n =?) must be indicated by gender and quartile.

When reading the document, there is no description of dietary supplements consumption. This should have been one of the exclusion criteria from the study or work as a confounding factor.

The discussion needs to be improved, comparisons of the study results with results from other studies described in the scientific literature should be made. Hypotheses must also be formulated to explain the different results for men and women.

Round 2

Reviewer 1 Report

The authors made a considerable effort to respond analytically to all comments.

The flowchart of exclusion criteria and appendix table 1 were important additions to the manuscript.

However, the referee suggests that Appendix Table 3 is not necessary to be included.

In addition, the following explanatory response on dietary data acquisition, should likely be incorporated in the main manuscript (as a whole or partially) because it reinforces the methodological background of the study:

Answer: Because the 24 h recall method is not an actual measurement, it is not the best method to assess nutrient intake precisely, as you noted. However, if taking measures to be representative of participants’ usual dietary habits excluding day-to-day variation and to reduce measurement variability depending on interviewers, it can be used as one of the reliable ways to analyze nutrients intake. Actually, some studies have shown that the 24 h recall method can be a sufficient method for assessing dietary intake after taking measures including the aforementioned (Navnit Kaur Grewal et al. Nutrients 2014;6:2333-2347). With this background, many nation-wide nutrition surveys including the National Health and Nutrition Examination Survey (NHANES) adopted the 24 h recall method as a reliable way to assess nutrient intake.

The only comment that has not been adequately addressed by the authors is that the majority of references are outdated (many being over 20years). In the revised manuscript the authors provided only one recent citation, but more should have been included in place of older ones, to reflect current knowledge.

Reviewer 2 Report

The authors have adequately responded to my comments, improving the manuscript which may be suitable for publication
